# Improving State-of-the-Art in One-Class Classification by Leveraging Unlabeled Data

## Abstract

Recent advances in One-Class (OC) classification combine the ability to learn exclusively from positive examples with the expressive power of deep neural networks. A cornerstone of OC methods is to make assumptions regarding negative distribution, e.g., that negative data are scattered uniformly or concentrated in the origin. An alternative approach employed in Positive-Unlabeled (PU) learning is to additionally leverage unlabeled data to approximate negative distribution more precisely. In this paper, our goal is to find the best ways to utilize unlabeled data on top of positive data in different settings. While it is reasonable to expect that PU algorithms outperform OC algorithms due to access to more data, we find that the opposite can be true if unlabeled data is unreliable, i.e. contain negative examples that are either too few or sampled from a different distribution. As an alternative to using existing PU algorithms, we propose to modify OC algorithms to incorporate unlabeled data. We find that such PU modifications can consistently benefit even from unreliable unlabeled data if they satisfy a crucial property: when unlabeled data consists exclusively of positive examples, the PU modification becomes equivalent to the original OC algorithm. Our main practical recommendation is to use state-of-the-art PU algorithms when unlabeled data is reliable and to use PU modifications of state-of-the-art OC algorithms that satisfy the formulated property otherwise. Additionally, we make a progress towards distinguishing the cases of reliable and unreliable unlabeled data using statistical tests.

## 1 Introduction

An input of a supervised binary classifier consists of two sets of examples: positive and negative. However, the access to clean samples from both classes can be obstructed in many realistic scenarios. A particularly well-studied restriction is the absence of clean negative examples. One of the approaches to deal with this restriction is One-Class (OC) classification (Moya et al., 1993). Typically, OC algorithms treat available positive examples as normal data and try to separate it from previously unseen data, referred to as anomalies or outliers (Grubbs, 1969; Hodge and Austin, 2004; Chandola et al., 2009; Chalapathy et al., 2018; Chalapathy and Chawla, 2019). As noted in other studies (Scott and Blanchard, 2009; Ruff et al., 2020a;b), OC algorithms necessarily make prior assumptions about negative distribution. For example, some methods assume negative distribution to be uniform (Vert et al., 2006; Scott and Nowak, 2006) or concentrated where positive data are rare (Tax and Duin, 2004; Ruff et al., 2018), model negative distribution as a Gaussian (Oza and Patel, 2018), or separate positive data from the origin (Schölkopf et al., 2000).

Another approach to classification in the absence of negative data is Positive-Unlabeled (PU) learning (Denis, 1998; Denis et al., 2005; Li and Liu, 2005). In addition to a clean positive sample, PU algorithms leverage an unlabeled set of mixed positive and negative examples. In contrast to OC methods, PU methods approximate negative distribution (Elkan and Noto, 2008; Ivanov, 2019), its statistics (Du Plessis et al., 2014; 2015; Kiryo et al., 2017), or samples from it (Yu et al., 2002; Liu et al., 2002; Li and Liu, 2003; Xu et al., 2019), by comparing positive and unlabeled data. This approach can even outperform supervised classification, given a sufficient amount of unlabeled data (Niu et al., 2016; Kiryo et al., 2017).

Because PU methods make fewer assumptions and have access to more data, they might seem favorable to OC methods whenever unlabeled data are at hand. However, our experiments show that

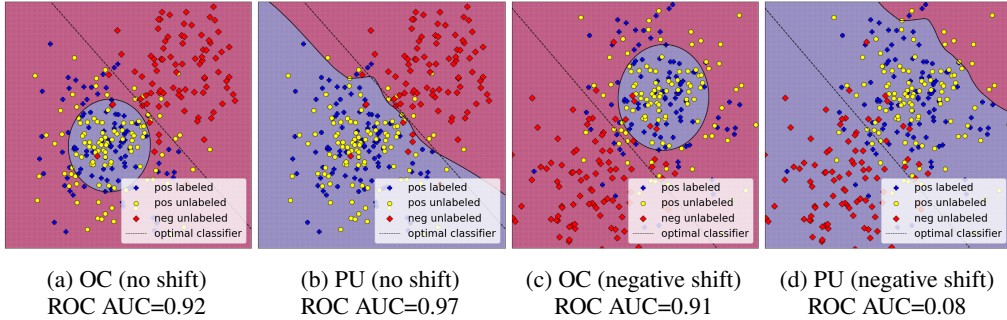

|  (a) OC (no shift)  |  (b) PU (no shift)  |  (c) OC (negative shift)  |  (d) PU (negative shift)  |
| --- | --- | --- | --- |
| ROC AUC=0.92 | ROC AUC=0.97 | ROC AUC=0.91 | ROC AUC=0.08 |

Figure 1: Performance of OC and PU models on synthetic data. Subfigures (a) and (b) show the performance of the algorithms in the standard case when negative distribution does not change from train to test. Subfigures (c) and (d) show the effects of negative distribution shift. The OC and PU models are OC-SVM and PU-SVM, respectively (section 3.2.1). The OC model is trained on labeled positive examples. The PU model is trained on labeled positive as well as unlabeled examples, i.e. a mixture of positive and negative examples. Black dashed lines represent the decision boundaries of the optimal Bayesian classifier. Overdependence on unreliable unlabeled data causes the PU method to misclassify all anomalies when a shift of negative distribution occurs.

dependence on unlabeled data may hinder PU methods in particularly extreme cases, which we refer to as cases of unreliable unlabeled data. We identify several such cases, including distributional shifts in unlabeled data, scarcity of unlabeled data, and scarcity of latent negative examples in unlabeled data. We present a motivational example of a possible effect of distributional shifts on PU models in Figure 1. In subplots (a, b) a PU model approximates the separating line more accurately than an OC model. Conversely, in subplots (c, d) a shift of negative distribution causes the PU model to misclassify all negative examples, whereas the OC model is unaffected by the shift.

One possible conclusion is to opt for using OC methods when unlabeled data is unreliable, but we aim to find a principled way to construct robust PU methods that can learn even from unreliable unlabeled data. To this end, we propose modifications of several modern OC algorithms that leverage unlabeled data. These modifications either are based on risk estimation techniques where the negative risk is approximated using positive and unlabeled data (Kiryo et al., 2017), or simply replace positive data with unlabeled in algorithm-specific routines (see Sections 3.2.4, 3.2.5). We find that all modifications benefit from reliable unlabeled data, but only modifications from the second group are safe to apply to unreliable unlabeled data (i.e. either improve upon or perform on par with the original OC methods). We pinpoint this result to a crucial property that only the modifications from the second group possess: in the absence of latent negative examples, the PU modification becomes equivalent to the original OC algorithm. Our main practical recommendation is to use state-of-the-art PU algorithms when reliable unlabeled data is available and to use modifications of state-of-the-art OC algorithms that satisfy the property when robustness is a concern.

A question remains how to identify the cases of unreliable unlabeled data. We find that scarcity of latent negatives can be statistically tested by comparing predictions of OC or PU models for positive and unlabeled samples. Similarly, a shift of negative distribution can be tested by comparing predictions for unlabeled data from training and testing distributions, providing the latter is available.

**Related Work** There is a line of work in the OC literature that investigates ways to augment OC methods with additional data. Several studies find that exposure to a small and possibly biased sample of outliers can improve the performance of OC classifiers (Hendrycks et al., 2018; Ruff et al., 2019; 2020b). Scott and Blanchard (2009) show both theoretically and empirically that unlabeled data help a classic machine learning algorithm to detect novelties. Nevertheless, there is a lack of modern literature that views OC and PU learning as different solutions to the same problem and gives practical recommendations for different scenarios. Our study attempts to fill this gap.

The problem of unreliable unlabeled data is also underexplored in PU literature. One existing direction is concerned with robustness to class prior shift (Charoenphakdee and Sugiyama, 2019). While this is a useful property, we study a more severe case of arbitrary negative distribution shifts.

Furthermore, an increase in proportions of positive class can be seen as a special case of negative distribution shift. We implement the latest PU method robust to class prior shifts as a baseline (Nakajima and Sugiyama, 2021). Other papers are concerned with arbitrary positive shifts (Hammoudeh and Lowd, 2020) and covariate shifts (Sakai and Shimizu, 2019), but unlike us, they assume access to unbiased unlabeled data from the testing distribution. To the best of our knowledge, we address the problem of PU learning with shifted or scarce latent negatives for the first time.

Our study is conceptually similar to the field of safe semi-supervised learning where the focus is on making unlabeled data never hurt while retaining performance (Li and Zhou, 2014; Li and Liang, 2019; Guo et al., 2020). The difference is that we do not assume access to labeled negative data.

## 2 PROBLEM SETUP

Let $x \in \mathbb{R}^d$ be a data point and let $y \in \{0, 1\}$ be a binary label. Let $s = 1$ if the data point is labeled and 0 otherwise. We assume that only positive data can be labeled, i.e. $p(s = 1|y = 0) = 0$. We view $x, y, s$ as random variables with some joint distribution $f(x, y, s)$. The probability density functions of positive or negative distributions are given by:

$$f_p(x) := f(x \mid y = 1, s = 0) \quad \text{(1a)} \qquad f_n(x) := f(x \mid y = 0, s = 0) \quad \text{(1b)}$$

Let $f_u(x)$ be the probability density function of unlabeled distribution, i.e. a mixture of positive and negative distributions, and $\alpha = p(y = 1|s = 0)$ be the mixture proportion. We assume that the probability of being labeled $p(s = 1 \mid y = 1)$ is constant and independent of $x$, which is known as Selected Completely At Random (SCAR) setting (Elkan and Noto, 2008). In this case:

$$f_u(x) := f(x|s = 0) = \alpha f_p(x) + (1 - \alpha) f_n(x) \tag{2}$$

In OC methods we assume that only a sample $X_p$ from $f_p(x)$ is available. In PU methods we assume the *case-control scenario* (Bekker and Davis, 2020): two sets of data $X_p$ and $X_u$ are sampled independently from $f_p(x)$ and $f_u(x)$. Both OC and PU methods output some score function proportional to $p(x) := f(y = 1|x, s = 0)$ that separates positive and negative data. PU methods often additionally estimate $\alpha$. Since $\alpha$ is generally unidentifiable (Blanchard et al., 2010), we will focus on estimation of its upper bound $\alpha^*$, as proposed in (Jain et al., 2016).

## 3 METHODS

In this section, we first describe PU methods used in this study. We begin by describing Risk Estimation approach, which some of our PU modifications of OC algorithms are based on. Then, we describe DRPU and PAN, which we use for comparison as the most recent and state-of-the-art PU algorithms. In Appendix A.5, we elaborate on this choice as state-of-the-art by comparing these algorithms with other existing PU algorithms. Finally, we describe OC methods used in this study and propose their PU modifications. Additional details are reported in Appendix B.

### 3.1 POSITIVE-UNLABELED METHODS

#### 3.1.1 RISK ESTIMATION

Let $h(x)$ be an arbitrary decision function that estimates $y$, $l(t, y)$ be loss function, i.e. the loss incurred for predicting $t$ when the ground truth is $y$. Define $R_p^+(h) = E_{x \sim f_p}(l(h(x), 1))$ and $R_n^-(h) = E_{x \sim f_n}(l(h(x), 0))$ as positive and negative risks. If both positive and negative data are available, the risk of the decision function can be estimated as a weighted sum of positive and negative risks (eq. 3a). In PU case, $R_n^-(h)$ is unavailable but can be estimated as a difference between risks on positive and unlabeled data (eq. 3b), as shown in (Du Plessis et al., 2014; 2015).

$$R_{pn}(h) = \alpha R_p^+(h) + (1-\alpha) R_n^-(h) \quad \text{(3a)} \qquad R_{pu}(h) = \alpha R_p^+(h) - \alpha R_p^-(h) + R_u^-(h) \quad \text{(3b)}$$

Estimator (3b) can be improved by introducing a non-negativity constraint to reduce overfitting (Kiryo et al., 2017):

$$R_{nn}(h) = \alpha R_p^+(h) + \max(0, -\alpha R_p^-(h) + R_u^-(h)) \tag{4}$$

Estimator (4) is called non-negative risk estimator. In practice, the decision function $h$ is parameterized by $\theta$, which can represent weights of a neural network or some other model. The parameters are trained to minimize $R_{nn}(h(x \mid \theta))$ for some loss function like double hinge (Du Plessis et al., 2015) or sigmoid (Kiryo et al., 2017). We use the latter. Notice that $\alpha$ is assumed to be identified in this method, so in experiments, we additionally estimate it with TIcE (Bekker and Davis, 2018).

Risk estimation can be applied to modify any OC model to leverage unlabeled data, providing this OC model is based on or can be generalized to the supervised (PN) setting. This can be done by first replacing the OC objective with the PN objective, and then applying risk estimation to the PN objective, i.e. evaluating negative risk using positive and unlabeled samples.

### 3.1.2 DRPU

DRPU (Nakajima and Sugiyama, 2021) is a state-of-the-art extension of Risk Estimation technique that relaxes the assumption of identified mixture proportions by leveraging density ratio estimation. Specifically, it minimizes the Bregman divergence between the posterior probability $p(x)$ and its parametric estimate. As a result, DRPU is stable to the shifts of class proportions in unlabeled data.

### 3.1.3 PAN

Predictive Adversarial Network (PAN) (Hu et al., 2021) is another state-of-the-art PU algorithm based on a GANs (Goodfellow et al., 2014). In this algorithm, the generator is replaced with a classifier trained to identify unlabeled examples that are likely to be considered positive by the discriminator. The discriminator then tries to maximize KL-divergence between its and the generator's predictions while correctly classifying labeled positive examples.

## 3.2 ONE-CLASS METHODS

### 3.2.1 OC-SVM

OC-SVM (Schölkopf et al., 2000) is one of the most well-known OC methods. OC-SVM modifies the classical objective of Support Vector Machines and tries to separate the nominal data from the origin. In the case of RBF kernel (and other translation-invariant kernels (Schölkopf et al., 2001)), OC-SVM tries to enclose positive data while minimizing enclosed volume and is equivalent to SVDD (David, 2001; Tax and Duin, 2004). Similarly to SVM, OC-SVM can be optimized by solving the dual problem and specifying only dot product in the feature space. OC-SVM solves the following optimization problem:

$$\min_{w,r} \frac{1}{2} \|w\|^2 - r + \frac{1}{\nu N} \sum_{i=0}^{N} \max(0, r - w \cdot \Phi(x_i)) - r \tag{5}$$

where $\Phi$ is a feature map in some Hilbert space, $\nu$ is both a regularization parameter and a correction for possible data contamination, and $w \cdot \Phi(x_i) - r$ is a decision function. Unlike most OC papers, we use linear kernel instead of RBF kernel in our experiments. During preliminary experiments, we discovered that such replacement improves ROC AUC of OC-SVM on benchmark datasets.

**PU**    As Du Plessis et al. (2015) show, the linear models can be trained in PU setting by minimizing risk defined in (3b). Since SVM is a linear model that minimizes empirical risk, one can use it as $h(x)$ and apply risk estimator techniques. In Appendix B.1.2, we show that similarly to SVM and OC-SVM it can be optimized by solving the dual problem, but that all labeled positive examples are support vectors in PU-SVM. As a faster alternative, we optimize non-negative risk (4) with Stochastic Gradient Decent (SGD), the details of which are reported in Appendix B.1.1.

### 3.2.2 OC-CNN

OC-CNN (Oza and Patel, 2018) uses a pretrained model like ResNet as a frozen feature extractor. In the feature space, a zero-centered Gaussian is used as the pseudo-negative distribution. After that, a supervised classifier is trained to distinguish positive and pseudo-negative examples. As in the original paper, we use ResNet pretrained on ImageNet as a feature extractor. Since pretraining is done in a supervised manner and classes from CIFAR10 overlap with classes from ImageNet, OC-CNN should not be directly compared with other OC algorithms that do not have access to such additional data. The same logic applies to the proposed PU modification.

**PU** Since OC-CNN assumes a particular negative distribution, it is easy to modify for PU setup. We replace the pseudo-negative examples in the latent space with unlabeled examples processed through the same feature extractor and then perform the standard risk estimation procedure, i.e. minimize non-negative risk in the latent space (4).

### 3.2.3 OC-LSTM

OC-LSTM (Ergen et al., 2017) combines Long Short-Term Memory (Hochreiter and Schmidhuber, 1997) and OC-SVM. First, the data points are processed through an LSTM block, after which OC-SVM is applied to the last hidden state of LSTM. One way to optimize such model is to freeze its components iteratively. Alternatively, OC-SVM can be optimized with SGD and trained end-to-end with LSTM. We employ the latter approach.

**PU** To adapt OC-LSTM for unlabeled data, we replace OC-SVM with our PU-SVM.

### 3.2.4 DROCC

DROCC (Goyal et al., 2020) achieves state-of-the-art performance on several real-world datasets across different domains. The key idea behind DROCC is that positive data lie on a low-dimensional locally Euclidean manifold. Based on this assumption, the pseudo-negative examples can be generated from available positives via gradient ascent, similarly to adversarial attacks. After that, a classifier is trained to minimize cross-entropy loss on positive and pseudo-negative examples.

**PU** In our PU modification of DROCC, we generate pseudo-negative points from unlabeled rather than positive examples. This way, the pseudo-negative examples are closer to the real negatives if unlabeled data contain some negative points, i.e. if $\alpha < 1$. In the extreme case when unlabeled data contain only positives, our modification is equivalent to the original algorithm.

### 3.2.5 CSI

CSI (Tack et al., 2020) is an OC method based on contrastive learning. In addition to contrasting a given example with other instances as in conventional contrastive learning, CSI contrasts the example with distributionally-shifted augmentations of itself, such as rotations or permutations of parts of an image. The original study uses ResNet as a neural architecture, but we train a small convolutional network instead for a fair comparison with other algorithms.

**PU** In our PU modification of CSI, we additionally contrast positive points with examples from unlabeled data. Like our PU-DROCC, such modification will be close to the original model in the extreme case when unlabeled data has no negative examples, i.e. $\alpha \sim 1$.

## 4 DATASETS

Here we briefly describe the datasets used in this study. For details, please refer to Appendix C.

**CIFAR-10** CIFAR-10 is a standard for both OC and PU methods benchmark dataset with images from ten animal and vehicle classes (Krizhevsky et al., 2009). There are around 6000 images for each class and the proportion of negatives and positives depends on the particular experiment.

**Abnormal1001** This dataset consists of abnormal images from six classes, including Chair, Car, Airplane, Boat, Sofa, and Motorbike (Saleh et al., 2013). Normal images come from the respective

classes from the PASCAL VOC dataset (Everingham et al., 2010). An example of images from this dataset is presented in Figure 2. We perform experiments only with Car class since it has the most examples, with 110 abnormal and 1315 normal images, $\alpha = 0.87$. This dataset is challenging for PU methods since the negative data are very scattered and only a few negative examples are available.

**Pendigits**  Pendigits dataset consists of consecutive pixels sampled from digits handwritten on a tablet (Asuncion and Newman, 2007). The task is to classify the digits based on these sequences of pixels. We use a processed dataset from ODDS with 6870 examples in total and 156 anomalies, $\alpha = 0.97$.

**Text Datasets**  We use two different datasets with text data. The first is a standard dataset of deceptive reviews (Ott et al., 2011). It consists of 400 deceptive hotel reviews acquired with crowdsourcing service and 400 truthful reviews from TripAdvisor, $\alpha = 0.33$. The second is SMS Spam dataset (Asuncion and Newman, 2007). It contains 5572 examples with 747 spam messages considered as anomalies, $\alpha = 0.76$.

## 5 EXPERIMENTS

In this section, we describe experimental settings and report results. We repeat each experiment 10 times. Statistical significance is verified via paired Wilcoxon signed-rank test with a 0.05 P-value threshold. We indicate a significant difference in favor of PU or OC method with a line under the higher ROC AUC value. As noted before (Section 3.2.2), the performance of OC-CNN and PU-CNN should not be directly compared to other models. The selected hyperparameters and tuning procedure are reported in Appendix D. Additional experiments are reported in Appendix A.

### 5.1 ONE-VS-ALL

A common experimental setting in OC papers for datasets with multiple classes is one-vs-all classification. In this setting, one class is treated as positive and all other classes constitute negative examples. We fix $\alpha$ in this setting as $0.5$, label half of the positive points, and conduct experiments with each class selected as positive. Since OC methods have access to only half of all positive points, their performance cannot be directly compared to the results reported in their original papers. ROC AUC metrics for OC and PU methods for all classes are presented in Table 1. PU modifications consistently significantly outperform their OC counterparts with few exceptions where methods perform on par. Furthermore, DRPU and PAN outperform other algorithms on most classes, although on classes 7 through 9 our PU-CSI performs exceptionally well.

Table 1: ROC AUC in one-vs-all setting

| Pos class | 0 | 1 | 2 | 3 | 4 | 5 | 6 | 7 | 8 | 9 | avg |
|---|---|---|---|---|---|---|---|---|---|---|---|
| OC-SVM | 0.78 | 0.67 | 0.60 | 0.67 | 0.63 | 0.64 | 0.74 | 0.66 | 0.68 | 0.75 | 0.68 |
| PU-SVM | 0.79 | 0.70 | 0.62 | 0.68 | 0.65 | 0.72 | 0.74 | 0.70 | 0.77 | 0.82 | 0.72 |
| CSI | 0.70 | 0.87 | 0.63 | 0.62 | 0.60 | **0.83** | 0.73 | 0.80 | 0.86 | 0.89 | 0.76 |
| PU-CSI | 0.71 | **0.90** | 0.61 | 0.68 | 0.62 | **0.83** | 0.79 | **0.92** | **0.89** | **0.91** | 0.79 |
| DROCC | 0.76 | 0.74 | 0.64 | 0.61 | 0.71 | 0.67 | 0.75 | 0.71 | 0.78 | 0.79 | 0.72 |
| PU-DROCC | 0.80 | 0.85 | 0.75 | 0.74 | 0.76 | 0.77 | 0.81 | 0.79 | 0.84 | 0.83 | 0.79 |
| OC-CNN | 0.74 | 0.80 | 0.64 | 0.69 | 0.77 | 0.76 | 0.81 | 0.73 | 0.78 | 0.83 | 0.76 |
| PU-CNN | 0.91 | 0.94 | 0.87 | 0.86 | 0.90 | 0.90 | 0.94 | 0.92 | 0.95 | 0.93 | 0.91 |
| DRPU | **0.85** | 0.89 | **0.78** | **0.78** | **0.81** | 0.82 | **0.88** | 0.83 | 0.88 | 0.87 | **0.84** |
| PAN | **0.85** | 0.89 | 0.76 | 0.77 | 0.80 | 0.80 | **0.88** | 0.82 | 0.88 | 0.87 | 0.83 |

### 5.2 SHIFT OF THE NEGATIVE DISTRIBUTION

While in the previous setting PU algorithms have shined, that setting is convenient in that unlabeled data contain reliable latent negative examples. In the next setting, we investigate the effect of negative distribution shift, i.e. when latent negative examples are sampled from different distributions

Table 2: ROC AUC for different modalities of negative distribution under random negative shifts on CIFAR-10

| Neg modality | 1 | 2 | 3 | 4 |
|---|---|---|---|---|
| OC-SVM | **0.76** | 0.78 | 0.80 | 0.748 |
| PU-SVM | 0.70 | 0.78 | 0.80 | 0.752 |
| CSI | 0.68 | 0.74 | 0.69 | 0.69 |
| PU-CSI | 0.73 | 0.76 | 0.70 | 0.70 |
| DROCC | 0.73 | 0.74 | 0.73 | 0.72 |
| PU-DROCC | 0.73 | 0.77 | 0.77 | 0.75 |
| OC-CNN | 0.72 | 0.71 | 0.74 | 0.74 |
| PU-CNN | 0.80 | 0.78 | 0.85 | 0.83 |
| DRPU | 0.74 | 0.77 | 0.80 | **0.81** |
| PAN | 0.74 | **0.79** | **0.81** | **0.81** |

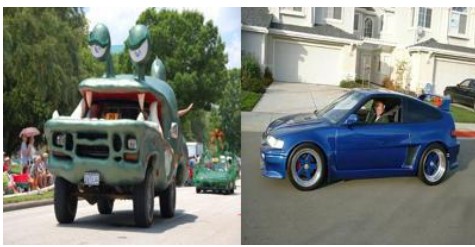

Figure 2: Examples of abnormal car (left) and normal car (right) from Abnormal1001.

at train and test times. For example, this can be relevant if the negative distribution constantly shifts over time and the model cannot be retrained at each time step, or if a large but biased unlabeled sample is available for training, whereas an unlabeled sample used for inference is either too small or for some reason unavailable. Since negative shift only affects unlabeled data, only PU methods are expected to be sensitive to it (Fig. 1). Further, we suspect that the effect of negative shift on PU methods might decrease with the increased modality of the negative distribution due to negative data covering more latent space, so we additionally vary the number of negative classes.

For this setting, we consider $0$ class from CIFAR-10 as positive, $n$ random classes as negative train, and $n$ other random classes as negative test, where $n \in \{1, 2, 3, 4\}$. The results are presented in Table 2. Despite the distribution shifts that negatively affect PU methods, most PU modifications either significantly outperform or perform on par with their OC counterparts. The exception is OC-SVM, which outperforms all other algorithms when $n = 1$. Evidently, even incorrect but data-driven estimates of the negative distribution used by PU algorithm are on average more helpful than the generic assumptions made by OC algorithms. Furthermore, for state-of-the-art PU algorithms increasing the number of negative classes partially mitigates the effect of negative shifts.

Although PU models still outperform their OC counterparts when the shift is random, there are particular shifts that can significantly harm the performance of PU algorithms. We present results of experiments with non-random negative shifts, such as animal-animal, animal-vehicle, vehicle-animal, and vehicle-vehicle, in Table 3. The results vary from algorithm to algorithm. OC-SVM and in some cases OC-CNN outperform their PU modifications, highlighting that modifications based on risk estimation are not robust to negative distribution shifts. PU-CSI shows the same performance as regular CSI, but both perform poorly compared to other models. This is likely due to CSI performing poorly on some classes chosen as positive (Table 1), as we fix the positive class as $0$ in the current setting. On the other hand, PU-DROCC significantly outperforms its OC counterpart in most cases and at least performs on par in other cases. It also outperforms both state-of-the-art PU algorithms in all cases but one. Indeed, this algorithm shows robustness to shifts in latent negative distribution. Experiments with a different class chosen as positive are reported in Appendix A.

## 5.3 SIZE AND CONTAMINATION OF UNLABELED DATA

Other cases when PU algorithms may struggle are when negative examples are scarce, i.e. when the positive class proportion is high or the sample size of unlabeled data is small. We study the performance of OC and PU methods with respect to $\alpha$ and the size of the unlabeled sample. We focus on state-of-the-art algorithms as well as some baselines, leaving other experiments to Appendix A. We choose all vehicles as positive classes and all animals as negative classes. The results are presented in Figure 3. Subplot (b) shows that even as few as 50 unlabeled examples are sufficient for PU models to outperform DROCC, but that both versions of CSI outperform other algorithms until the unlabeled sample becomes bigger than 1000 examples. Subplot (a) shows that state-of-the-art PU methods struggle when unlabeled examples are mostly positive. In contrast, PU-DROCC consistently outperforms its OC analog. While both versions of CSI perform similarly, they outperform all

Table 3: ROC AUC for particular shifts with Airplane class (0) chosen as positive. We test shifts for unimodal and multimodal negative distributions. We consider following multimodal negative distributions: $A_1 = \{2, 3, 4\}$, $A_2 = \{5, 6, 7\}$, and $V_1 = \{1, 8, 9\}$

| Shift | $1 \to 2$ | $2 \to 1$ | $1 \to 8$ | $2 \to 3$ | $V_1 \to A_1$ | $A_1 \to V_1$ | $A_1 \to A_2$ |
|---|---|---|---|---|---|---|---|
| OC-SVM | **0.80** | 0.76 | 0.51 | **0.86** | 0.84 | 0.64 | 0.86 |
| PU-SVM | 0.77 | 0.63 | 0.55 | 0.79 | 0.78 | 0.57 | 0.85 |
| CSI | 0.75 | 0.54 | 0.46 | 0.54 | 0.75 | 0.49 | 0.80 |
| PU-CSI | 0.76 | 0.56 | 0.47 | 0.56 | 0.74 | 0.49 | 0.79 |
| DROCC | 0.74 | 0.76 | 0.60 | 0.80 | 0.75 | 0.68 | 0.79 |
| PU-DROCC | 0.75 | **0.79** | 0.66 | 0.85 | 0.77 | **0.69** | **0.87** |
| OC-CNN | 0.80 | 0.60 | 0.50 | 0.80 | 0.81 | 0.56 | 0.82 |
| PU-CNN | 0.63 | 0.66 | 0.65 | 0.92 | 0.69 | 0.71 | 0.96 |
| DRPU | 0.63 | 0.59 | **0.67** | 0.73 | 0.92 | 0.59 | 0.61 |
| PAN | 0.61 | 0.58 | 0.66 | 0.72 | **0.93** | 0.62 | 0.63 |

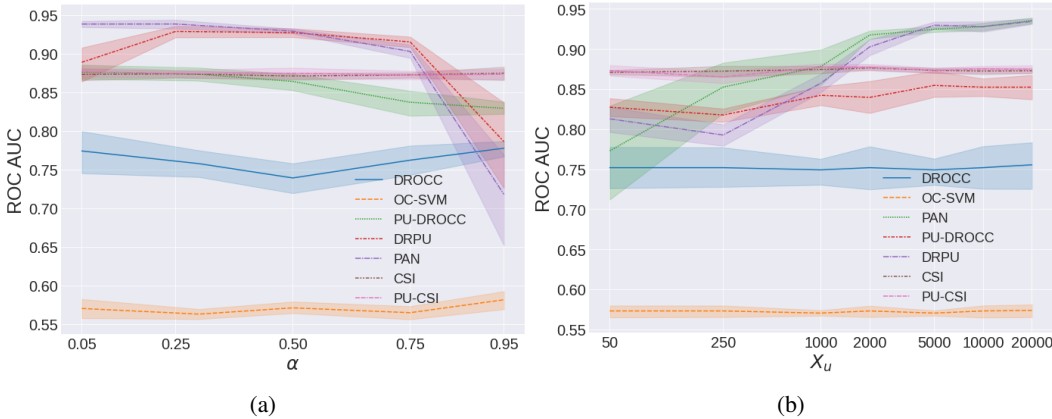

(a)                                            (b)

Figure 3: ROC AUC with respect to the proportion of positive examples in unlabeled data (a) and the size of unlabeled data (b).

other algorithms when $\alpha$ is high. Note that when $\alpha = 0.05$, the difference between CSI and PU-CSI is statistically significant in favor of the latter. Also note that CSI generally performs much better than in the previous settings, as its performance varies with the selected positive classes. As a final note, we investigate the effect of modality of positive data in more detail in Appendix A.1.

### 5.4 ABNORMAL1001

The results of the experiments on Abnormal are presented in Table 4. Although this dataset is challenging for PU methods due to the scarcity of anomalies and similarity of nominal and anomalous data, PU modifications still outperform OC counterparts across all models. Furthermore, PAN and our PU-DROCC perform the best among all methods.

Table 4: ROC AUC on Abnormal1001

| Base | OC | PU |
|---|---|---|
| SVM | 0.60 | 0.64 |
| CSI | 0.55 | 0.64 |
| DROCC | 0.60 | **0.84** |
| CNN | 0.53 | 0.76 |
| DRPU | – | 0.78 |
| PAN | – | **0.84** |

Table 5: ROC AUC on sequential data

| Model | PenDigits | Deceptive | SMS Spam |
|---|---|---|---|
| OC-LSTM | 0.96 | 0.52 | 0.69 |
| PU-LSTM | **0.97** | **0.77** | **0.92** |

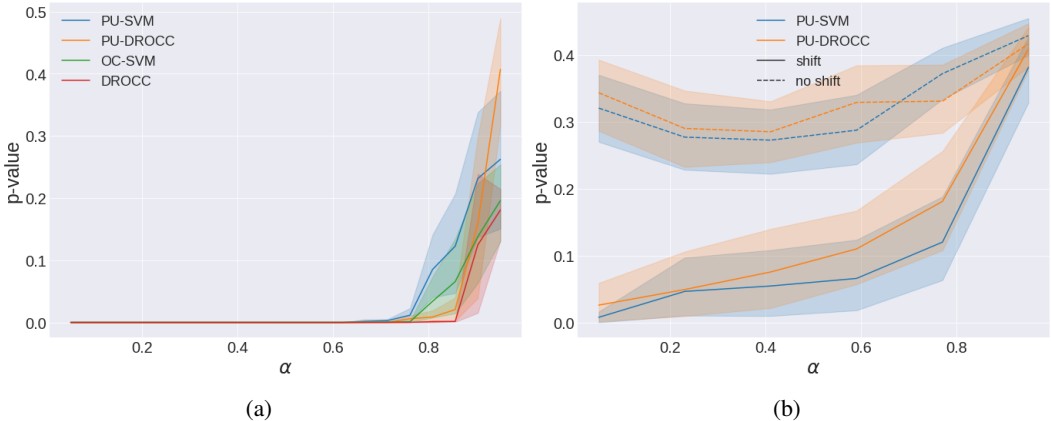

(a)                                    (b)

Figure 4: P-values of Mann–Whitney U-test for identification of a) high $\alpha$ or b) negative shift. We show results for OC-SVM and DROCC as OC models and PU-SVM and PU-DROCC as PU models.

## 5.5 SEQUENTIAL DATA

Comparison of LSTM-based models on datasets with sequential data can be found in Table 5. PU models achieve better scores than OC algorithms on all three datasets. Furthermore, OC-LSTM performs poorly on the Deceptive Reviews dataset, which has very few labeled positive examples.

## 5.6 IDENTIFYING UNRELIABLE UNLABELED DATA

As our experiments show, some PU algorithms can perform poorly when available unlabeled data are unreliable. Here, we propose a method to distinguish such cases. Empirically, PU models can struggle when $\alpha$ is too high ($\alpha > 0.9$), when negative shifts occur, or when only a few unlabeled examples are at hand. We discover that statistical tests can help with the first two problems. Applying Mann-Whitney U-test to the outputs of OC or PU models for unlabeled and positive samples can identify if $\alpha$ is high (fig. 4 a). Similarly, applying Mann-Whitney U-test on outputs of PU models for train and test unlabeled samples can help to detect distribution shifts. Fig. 4 (b) shows p-values for samples with and without negative shifts, which only overlap when $\alpha$ is high.

## 6 CONCLUSION

In this paper, we propose new PU algorithms based on existing OC algorithms, compare these and state-of-the-art PU algorithms in different settings, and formulate a general guideline for constructing a PU algorithm robust to unreliable unlabeled data, i.e. that it should become an OC algorithm when unlabeled data are close to being purely positive. We find that our algorithms that follow this guideline never underperform their OC counterparts, but that safety can come at a cost as PU SO-TAs generally outperform our algorithms when unlabeled data are reliable. Furthermore, we find that our PU modifications based on risk estimation consistently improve upon OC algorithms when unlabeled data are reliable. This approach is applicable to any OC algorithm that has a supervised analog and therefore presents another principled way to leverage unlabeled data that does not require drastic changes of the algorithm, albeit does not guarantee improvement if unlabeled data are unreliable. We hope that our findings will motivate future researchers to investigate augmentations of their OC algorithms that leverage unlabeled data, possibly by following our guideline.

On the other hand, to the best of our knowledge, our paper is the first to explicitly formulate the cases of unreliable unlabeled data and to investigate the potential vulnerability of PU algorithms to such cases. While we make a progress towards identifying such cases via statistical tests, a more desirable alternative is a PU algorithm that achieves state-of-the-art performance regardless of whether unlabeled data are reliable. We hope that future researchers will consider testing the robustness of their PU algorithms in the settings of unreliable unlabeled data, as well as proposing safe alternatives to their PU algorithms that should be in such settings.

## REPRODUCIBILITY STATEMENT

Details about all studied datasets can be found in Section 4 and Appendix C. In supplementary material, we provide code for data preprocessing, as well as all for all methods and experiments described in the paper. Hyperparameters and their tuning procedure are reported in Appendix D.

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

# A ADDITIONAL EXPERIMENTS

## A.1 NUMBER OF POSITIVE MODES

In the standard one-vs-all setting the positive distribution is close to unimodal. Many OC methods try to envelope positive data and thus greatly benefit from unimodality (Ghafoori and Leckie, 2020). However, distributions of real-world data are often more complex. Therefore, the one-vs-all setting might produce overly optimistic estimates of the performance of OC methods. We present a synthetic example that highlights the potential struggle of OC methods with multimodal positive distributions in the case when the negative points are concentrated between the modes of the positive distribution (Fig. 6 a). In this example, the OC method attempts to enclose all positive modes in a single envelope and assigns high probability of being positive to negative examples. In contrast, PU methods are robust to multimodality and can accurately estimate the decision boundary (Fig. 6 b).

In this setting, we consider a random subset from vehicle (animal) CIFAR-10 classes as positive data and all animal (vehicle) classes as negative data. We study the performance of models with respect to the number classes that form a positive sample. The results are reported in Tables 6, 7. Like in the one-vs-all setting, PU modifications consistently outperform their OC counterparts. Further, the performance of all OC methods decreases with the increased number of modes. OC-SVM suffers from multimodality the most, which is expected from a method that tries to enclose all positive data in a single envelope. The performance of some PU modifications also drops due to data becoming more complex, but these drops are slight compared to OC algorithms. The exception is PU-CSI that performs exactly as well as the original CSI on this task. Interestingly, the performance of SOTA PU methods generally increases with the modality. As a result, DRPU and PAN outperform all other algorithms when the modality is high (as in other experiments, we exclude CNN-based models from this comparison due to leveraging a ResNet pretrained on ImageNet). However, when the modality is low, PU-DROCC achieves even better performance.

Table 6: ROC AUC for different modalities of positive distribution on CIFAR-10 with vehicle classes chosen as positive

| Pos modality | 1 | 2 | 3 | 4 |
|---|---|---|---|---|
| OC-SVM | 0.74 | 0.60 | 0.58 | 0.57 |
| PU-SVM | 0.80 | 0.80 | 0.77 | 0.80 |
| CSI | 0.85 | 0.82 | 0.83 | 0.88 |
| PU-CSI | 0.85 | 0.82 | 0.83 | 0.88 |
| DROCC | 0.78 | 0.79 | 0.77 | 0.73 |
| PU-DROCC | **0.92** | **0.87** | 0.86 | 0.83 |
| OC-CNN | 0.90 | 0.90 | 0.81 | 0.82 |
| PU-CNN | 0.98 | 0.98 | 0.96 | 0.96 |
| DRPU | 0.88 | 0.86 | **0.88** | **0.93** |
| PAN | 0.87 | 0.85 | **0.88** | **0.93** |

Table 7: ROC AUC for different modalities of positive distribution on CIFAR-10 with animal classes chosen as positive

| Pos modality | 1 | 2 | 3 | 4 | 5 | 6 |
|---|---|---|---|---|---|---|
| OC-SVM | 0.60 | 0.64 | 0.69 | 0.66 | 0.64 | 0.67 |
| PU-SVM | 0.63 | 0.65 | 0.72 | 0.68 | 0.66 | 0.69 |
| CSI | 0.64 | 0.60 | 0.59 | 0.55 | 0.53 | 0.51 |
| PU-CSI | 0.66 | 0.60 | 0.59 | 0.53 | 0.52 | 0.51 |
| DROCC | 0.73 | 0.75 | 0.74 | 0.69 | 0.72 | 0.69 |
| PU-DROCC | **0.90** | **0.91** | **0.90** | **0.89** | **0.89** | 0.90 |
| OC-CNN | 0.76 | 0.81 | 0.70 | 0.78 | 0.74 | 0.74 |
| PU-CNN | 0.97 | 0.97 | 0.95 | 0.97 | 0.96 | 0.96 |
| DRPU | 0.82 | 0.78 | 0.77 | 0.82 | 0.86 | **0.93** |
| PAN | 0.82 | 0.77 | 0.77 | 0.81 | 0.85 | **0.93** |

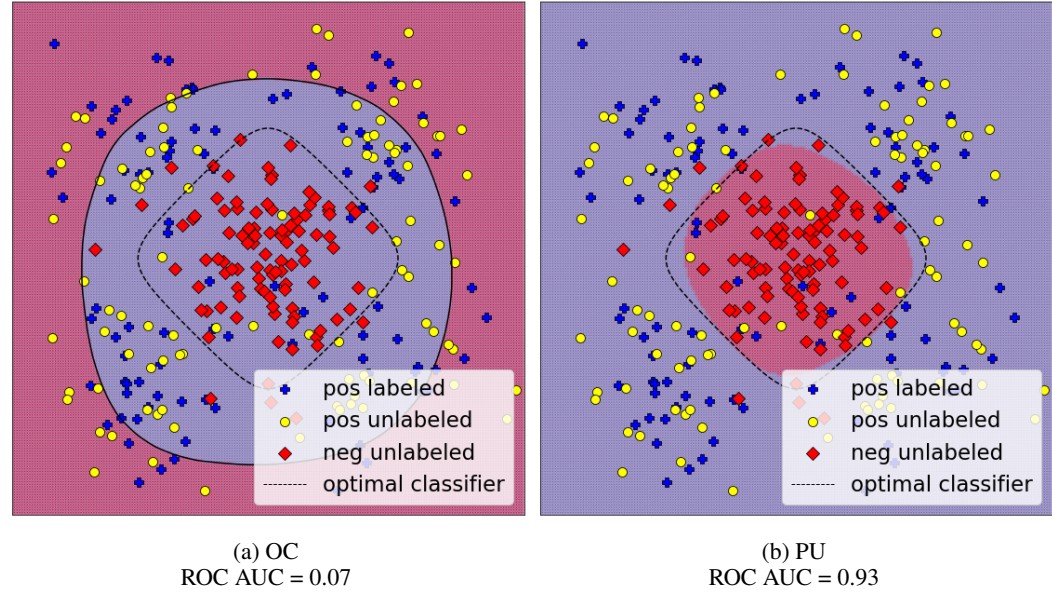

(a) OC
ROC AUC = 0.07

(b) PU
ROC AUC = 0.93

Figure 6: Separating lines of OC and PU algorithms on multimodal synthetic data. Positive distribution is a mixture of Gaussians with four different centers. The OC and PU models are OC-SVM and PU-SVM, respectively. OC-SVM tries to enclose all modes in a single envelope, whereas PU-SVM correctly identifies the location of negative data.

Table 8: ROC AUC for different modalities of negative distribution under random negative shifts on CIFAR-10 with Bird (2) class chosen as positive

| Neg modality | 1 | 2 | 3 | 4 |
|---|---|---|---|---|
| OC-SVM | 0.54 | 0.59 | 0.62 | 0.61 |
| PU-SVM | 0.54 | 0.61 | 0.64 | 0.60 |
| CSI | 0.59 | 0.61 | _0.58_ | 0.60 |
| PU-CSI | 0.54 | 0.59 | 0.56 | 0.57 |
| DROCC | 0.69 | 0.68 | 0.64 | 0.64 |
| PU-DROCC | **0.72** | **0.73** | **0.71** | _0.70_ |
| OC-CNN | 0.63 | 0.58 | 0.58 | 0.64 |
| PU-CNN | 0.60 | _0.73_ | _0.76_ | _0.81_ |
| DRPU | **0.72** | 0.70 | 0.70 | **0.72** |
| PAN | 0.64 | 0.65 | 0.70 | 0.69 |

## A.2 SHIFT OF THE NEGATIVE DISTRIBUTION

Table 8 and 9 show results of experiments under random and specific negative distribution shifts with bird class chosen as positive. The results are similar to those in the main text. These experiments show that while some shifts are not too detrimental for PU methods, there exist particular bad shifts that are. We also see that performance of OC methods varies drastically depending on the negative class, meaning that some classes are more difficult to distinguish than others regardless of shifts. Still, on average PU modifications can provide significant improvements. Moreover, our PU-DROCC is robust to shifts as it consistently outperforms the original DROCC and also outperforms on average other methdods, including PU SOTAs, in cases when modality of negative distribution is not too high. Finally, note that we consider rather drastic variants of negative shifts. In the real world, distribution might shift over time more smoothly. It also can be possible for PU methods to adapt to shifts by incorporating new reliable unlabeled data into training.

Table 9: ROC AUC for particular shifts with Bird class (2) chosen as positive. We test shifts for unimodal and multimodal negative distribution. We consider following multimodal negative distributions: $A_3 = \{3, 4, 5\}$, $A_4 = \{6, 7\}$, $V_2 = \{0, 1\}$, $V_3 = \{8, 9\}$

| Shift | $0 \to 3$ | $3 \to 0$ | $0 \to 1$ | $3 \to 4$ | $V_2 \to A_3$ | $A_3 \to V_2$ | $V_2 \to V_3$ | $A_3 \to A_4$ |
|---|---|---|---|---|---|---|---|---|
| OC-SVM | 0.53 | **0.77** | 0.66 | 0.47 | 0.75 | 0.47 | 0.52 | 0.72 |
| PU-SVM | 0.68 | 0.59 | 0.72 | 0.47 | 0.68 | 0.57 | 0.60 | 0.66 |
| CSI | **0.70** | 0.52 | 0.66 | **0.63** | 0.66 | 0.59 | 0.51 | 0.59 |
| PU-CSI | 0.69 | 0.55 | 0.71 | 0.60 | 0.69 | 0.60 | 0.50 | 0.58 |
| DROCC | 0.67 | 0.70 | 0.78 | 0.53 | 0.73 | 0.63 | 0.60 | 0.72 |
| PU-DROCC | 0.63 | 0.69 | 0.80 | 0.58 | **0.88** | **0.67** | 0.59 | 0.68 |
| OC-CNN | 0.61 | 0.77 | 0.66 | 0.55 | 0.78 | 0.56 | 0.57 | 0.71 |
| PU-CNN | 0.58 | 0.58 | 0.83 | 0.64 | 0.97 | 0.79 | 0.58 | 0.71 |
| DRPU | 0.53 | 0.54 | **0.89** | 0.54 | 0.51 | 0.62 | **0.91** | **0.73** |
| PAN | 0.53 | 0.54 | 0.83 | 0.54 | 0.52 | 0.62 | **0.91** | **0.73** |

### A.3 SIZE AND CONTAMINATION OF UNLABELED DATA

In the main text, we report performance of the few selected OC and PU methods with respect to the proportions $\alpha$ and the size of unlabeled data $|X_u|$. Here, we additionally report results for all studied OC models and their PU modifications. The results are presented in Figures 7 and 8. Figure 7 shows that PU modifications usually outperform original models or perform on par, even when $\alpha$ is particularly high. The result, however, is compounded with multimodality of positive distribution, which might have a negative effect on some OC models (see Appendix A.1). Figure 8 shows that for most models as few unlabeled examples as 250 are enough to significantly improve performance. Moreover, PU modifications of SVM and PU-DROCC always outperform their OC analogues, even when unlabeled sample is extremely small.

### A.4 TWITTER BOTS

Results for experiments on Twitter dataset shown in fig. 10. For the description of the dataset, see Appendix C. Since the dataset is tabular, it is difficult to apply CNN-based and CSI-based models, so we only apply the rest. In general, we find that PU modifications outperform original models, especially in a more realistic case when new bots are added to the already existing. In most cases, state-of-the-art PU models achieve the best performance. It is worth noting that the results of (Rodríguez-Ruiz et al., 2020) for OC-SVM model are different for some classes. We have discovered that with some seeds it is possible to replicate results of the original paper, and that the authors appear to have used one random seed in their experiments. However, when results are averaged by seed, the quality of OC-SVM model can drop significantly. Another possible explanation is the slight differences in the datasets that we and Rodríguez-Ruiz et al. (2020) use.

Table 10: ROC AUC for different negative test distributions on Twitter dataset.

| Negatives | all | social1 | social2 | social3 | traditional1 |
|---|---|---|---|---|---|
| OC-SVM | 0.61 | 0.98 | 0.57 | 0.90 | 0.20 |
| PU-SVM | 0.86 | 0.71 | 0.89 | 0.77 | **0.98** |
| DROCC | 0.72 | 0.93 | 0.74 | 0.95 | 0.28 |
| PU-DROCC | 0.81 | 0.96 | 0.88 | 0.96 | 0.28 |
| DRPU | **0.94** | 0.94 | **0.96** | 0.93 | 0.89 |
| PAN | 0.83 | **0.99** | 0.87 | **0.99** | 0.45 |

### A.5 PU MODELS

Some modern PU methods moved from ROC AUC as the main metric to accuracy. Therefore, it is hard to compare different methods since in each paper they are optimized for their respective metrics. In this experiment, we try to perform a fair comparison between modern and classic PU

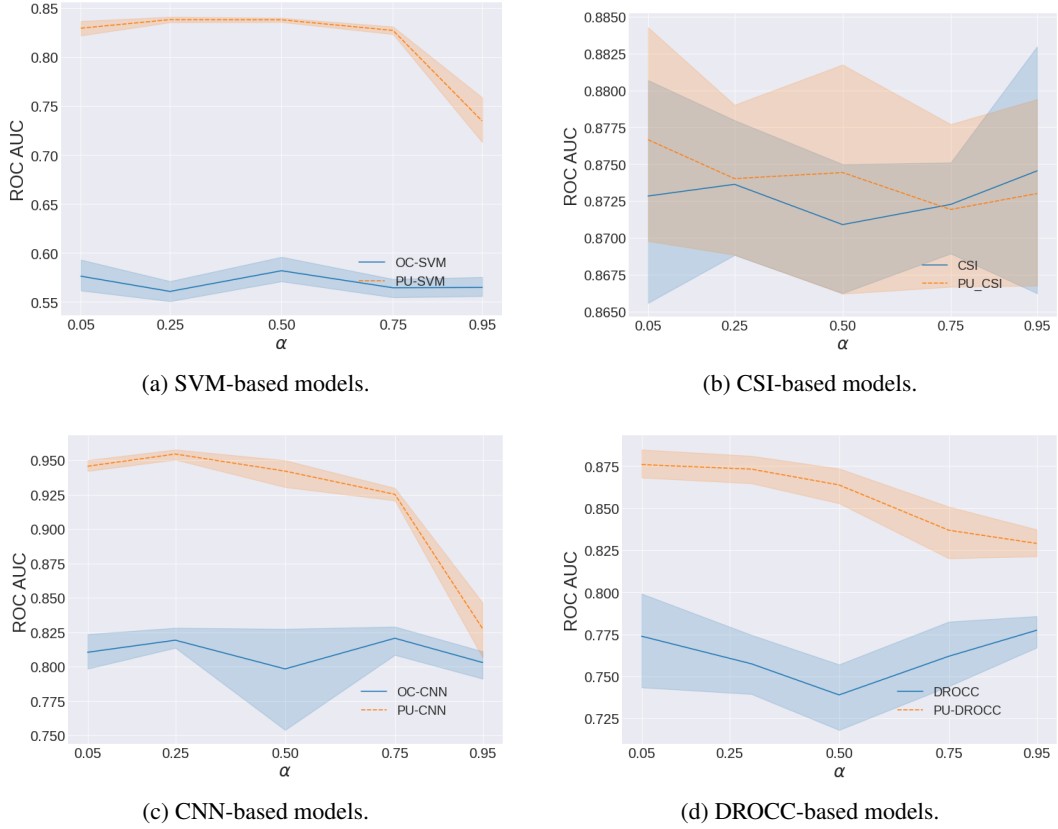

Figure 7: ROC AUC with respect to proportion of positive examples in unlabeled data.

Table 11: ROC AUC of PU models for various $\alpha$

| $\alpha$ | 0.05 | 0.25 | 0.5 | 0.75 | 0.95 |
|---|---|---|---|---|---|
| VPU | **0.95** | **0.94** | 0.92 | 0.86 | 0.58 |
| DRPU | 0.90 | 0.93 | **0.93** | **0.91** | **0.81** |
| PAN | **0.95** | **0.94** | **0.93** | **0.91** | 0.69 |
| EN | 0.94 | 0.93 | **0.93** | 0.88 | 0.66 |
| nnPU | 0.64 | 0.74 | 0.91 | **0.91** | 0.78 |
| nnPU* | 0.91 | 0.92 | **0.93** | 0.85 | 0.72 |
| DEDPUL | 0.94 | 0.93 | 0.92 | 0.88 | 0.68 |

methods. We choose vehicle classes from CIFAR-10 as positive and animal classes as negatives. The results are reported in Table 11. All methods are described in Appendix B.7 and Section 3.1. In Table 11, nnPU row represent nnPU with $\alpha$ estimated from DEDPUL, and nnPU* uses real value of $\alpha$. It can be seen that DRPU outperforms other models when $\alpha <= 0.75$, while PAN outperforms other models when $\alpha >= 0.5$. Since these two models cover the best performance in all cases, we use them as state-of-the-art for comparison in the main text.

## A.6 OC MODELS

We perform additional comparison with HRN, another OC model that was published concurrently with CSI. The results are reported in Table 12. Despite using the code provided by the authors, HRN performed poorly in our experiments and significantly differs from the results reported in the original paper (reported as HRN*). Our hypothesis to why this could have happened is that, according to the code, the authors have reported the highest ROC-AUC achieved during training, while we evaluate ROC-AUC after the training is complete for all our methods. Still, even if we

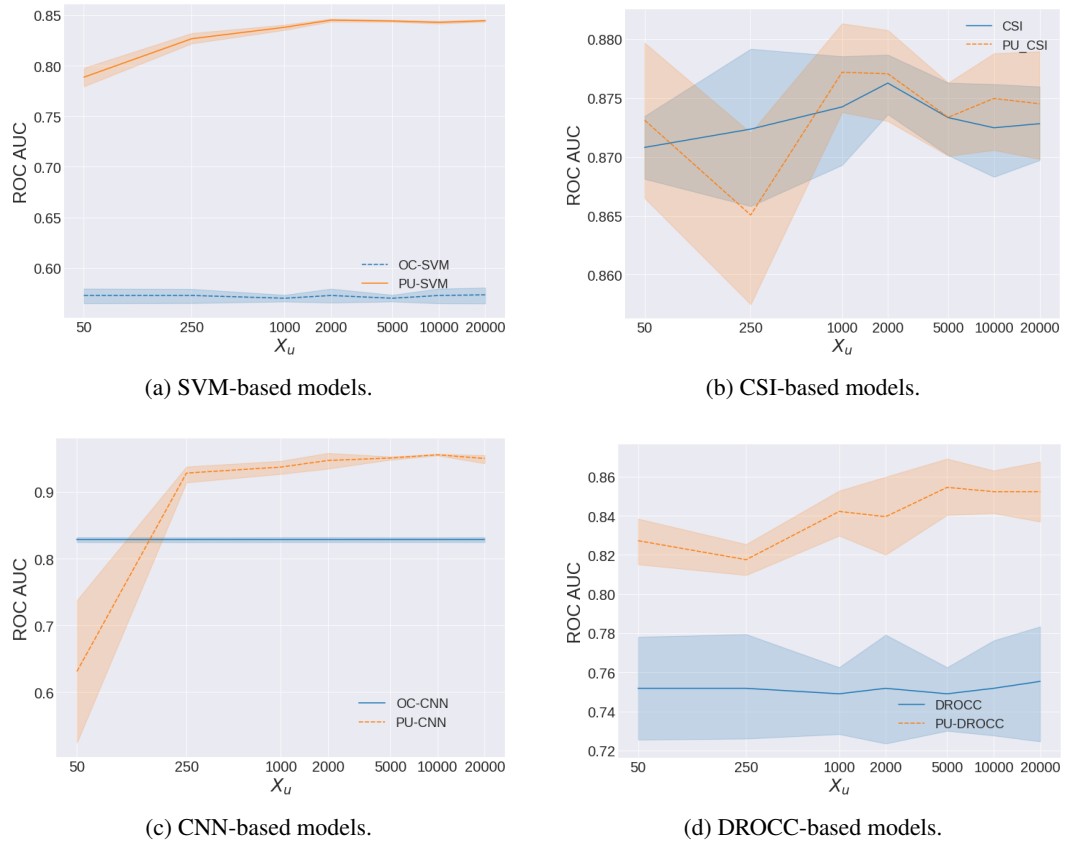

Figure 8: ROC AUC for different sizes of available unlabeled data.

use the results reported in the original paper for comparison, CSI and DROCC generally outperform HRN, so we do not include it for comparison in the main text.

Table 12: ROC AUC in one-vs-all setting for OC models

| Pos class | 0 | 1 | 2 | 3 | 4 | 5 | 6 | 7 | 8 | 9 |
|-----------|------|------|------|------|------|------|------|------|------|------|
| OC-SVM | **0.78** | 0.67 | 0.60 | **0.67** | 0.63 | 0.64 | 0.74 | 0.66 | 0.68 | 0.75 |
| CSI | 0.70 | **0.87** | 0.63 | 0.62 | 0.60 | **0.83** | 0.73 | **0.80** | **0.86** | **0.89** |
| DROCC | 0.76 | 0.74 | **0.64** | 0.61 | **0.71** | 0.67 | **0.75** | 0.71 | 0.78 | 0.79 |
| HRN | 0.74 | 0.5 | 0.5 | 0.5 | 0.5 | 0.5 | 0.5 | 0.5 | 0.67 | 0.55 |
| HRN* | 0.73 | 0.69 | 0.57 | 0.63 | 0.71 | 0.67 | 0.77 | 0.65 | 0.78 | 0.77 |

# B  METHODS

## B.1  PU-SVM

This subsection is organized as follows. First, we construct PU-SVM from classic SVM model and describe our implementation that is based on optimization with SGD. Second, we prove that PU-SVM without non-negativity constraint can also be solved via dual problem, albeit less efficiently.

### B.1.1  OUR IMPLEMENTATION

The soft-margin SVM objective can be formulated in the following way:

$$\min_{\omega,b} \lambda \|w\|^2 + E_{(x,y)\sim f_u} l_h(y, w \cdot \Phi(x) - b) \tag{6}$$

where $l_h(y,x) = max(0, 1 - yx)$ is hinge loss, $\lambda$ is regularization parameter, and $\omega \cdot \Phi(x) - b$ is the decision function. Since the objective (6) is a risk with regularization term, we can apply risk estimation techniques to it. However, several changes are required to that end. First, we replace hinge loss with double hinge loss $l_{dh}(y,x) = \max(-2yx, l_h(y,x))$. As shown in (Du Plessis et al., 2015), the unbiased estimate of the objective (6) with double hinge-loss is a convex function (7).

$$\min_{\omega,b} \lambda \|\omega\|^2 + E_{x_u \sim f_u} l_{dh}(-1, \omega\Phi(x_u) - b) +$$
$$+ \alpha E_{x_p \sim f_p} l_{dh}(1, \omega\Phi(x_p) - b) - \alpha E_{x_p \sim f_p} l_{dh}(-1, \omega\Phi(x_p) - b) \tag{7}$$

As we show in Section B.1.2, the objective (7) can be solved via dual problem, but the solution is computationally inefficient. As an alternative, we add an additional non-negative constraint to (7) and solve the resulting objective with the stochastic gradient decent (SGD):

$$\min_{\omega,b} \lambda \|\omega\|^2 + \alpha E_{x_p \sim f_p} l_{dh}(1, \omega\Phi(x_p) - b) +$$
$$+ \max(0, E_{x_u \sim f_u} l_{dh}(-1, \omega\Phi(x_u) - b) - \alpha E_{x_p \sim f_p} l_{dh}(-1, \omega\Phi(x_p) - b)) \tag{8}$$

The non-negativity constraint is motivated by (Kiryo et al., 2017). Since we solve (8) with SGD, we need to explicitly construct feature space for $x$, i.e. $\Phi(x)$, which again is inefficient. Instead, we make our second change and replace the dot-product in the feature space $w \cdot \Phi(x)$ with the kernel function $K(w, x)$. Note that now $w$ has same dimension as $x$, rather than the dimension of the feature space $\Phi(x)$. Similar trick is applied in OC-NN (Chalapathy et al., 2018) and DeepSVDD (Ghafoori and Leckie, 2020), where the dot-product is replaced with inference of neural network. Finally, we obtain the PU-SVM objective that we use in our study:

$$\min_{\omega,b} \lambda \|\omega\|^2 + \alpha E_{x_p \sim f_p} l_{dh}(1, K(\omega, x) - b) +$$
$$+ \max(0, E_{x_u \sim f_u} l_{dh}(-1, K(\omega, x) - b) - \alpha E_{x_p \sim f_p} l_{dh}(-1, K(\omega, x) - b)) \tag{9}$$

### B.1.2 DUAL PROBLEM

Here we show how the objective (7) can be solved via dual problem. To this end, we use the following property of double-hinge loss:

$$l_{dh}(1, x) - l_{dh}(-1, x) = \max(-2x, 0, 1 - x) - \max(2x, 0, 1 + x) = -2x \tag{10}$$

After replacing the expectations in (7) with their empirical estimates and applying the property (10), we get the following optimization problem:

$$\min_{\omega,b} L(\omega, b) = \min_{\omega,b} \left( \lambda \|\omega\|^2 + \frac{1}{n_u} \sum_{i \in u} l_{dh}(-1, \omega\Phi(x_i) - b) + 2\alpha b + \frac{2\alpha}{n_p} \sum_{i \in p} -\omega\Phi(x_i) \right) \tag{11}$$

Similarly to classic SVM, for each unlabeled point we introduce an additional slack variable $\xi_i = l_{dh}(-1, \omega\Phi(x_i) - b)$, such that:

$$\xi_i \geq 0 \tag{12}$$
$$\xi_i \geq 1 - b + \omega\Phi(x_i) \tag{13}$$
$$\xi_i \geq 2(\omega\Phi(x_i) - b) \tag{14}$$

Or, in another form:

$$-\xi_i \leq 0 \tag{15}$$

$$1 - \xi_i - b + \omega\Phi(x_i) \leq 0 \tag{16}$$

$$-\xi_i - 2(b - \omega\Phi(x_i)) \leq 0 \tag{17}$$

According to Karush–Kuhn–Tucker conditions (Karush, 1939; Kuhn and Tucker, 2014), the optimization problem (11) can be solved with the following Lagrangian function:

$$\mathfrak{L}(\omega, b, \xi, A, B, C) = \lambda \left\|\omega\right\|^2 - \frac{\alpha}{n_p} w \sum_{i \in p} \Phi(x_i) + 2\alpha b + \frac{1}{n_u} \sum_{i \in u} \xi_i - \sum_{i \in u} A_i \xi_i$$

$$- \sum_{i \in u} B_i(b - \omega\Phi(x_i) - 1 + \xi_i) - \sum_{i \in u} C_i(\xi_i - 2(\omega\Phi(x_i) - b)) \tag{18}$$

As KKT suggests, the optimal vector for the problem above satisfies the following conditions:

$$A_i \xi_i = 0 \tag{19}$$

$$B_i(b - \omega\Phi(x_i) - 1 + \xi_i) = 0 \tag{20}$$

$$C_i(\xi_i - 2(\omega\Phi(x_i) - b)) = 0 \tag{21}$$

$$\frac{\partial\mathfrak{L}}{\partial\omega} = \frac{\partial\mathfrak{L}}{\partial b} = \frac{\partial\mathfrak{L}}{\partial\xi_i} = 0 \tag{22}$$

Tacking a closer look at each derivative yields:

$$\frac{\partial\mathfrak{L}}{\partial\omega} = 2\omega - \frac{\alpha}{n_p} \sum_{i \in p} \Phi(x_i) + \sum_{i \in u}(B_i + 2C_i)\Phi(x_i) \tag{23}$$

$$\frac{\partial\mathfrak{L}}{\partial b} = 2\alpha - \sum_{i \in u} B_i - 2 \sum_{i \in u} C_i \tag{24}$$

$$\frac{\partial\mathfrak{L}}{\partial\xi_i} = \frac{1}{n_u} - A_i - B_i - C_i \tag{25}$$

Define $\tau_i$ as:

$$\tau_i = \begin{cases} \frac{\alpha}{2n_p} & , i \in p \\ -\frac{1}{2}B_i - C_i & , i \in u \end{cases} \tag{26}$$

Then, $\omega$ can be rewritten as:

$$\omega = \sum_{i \in p,u} \tau_i \Phi(x_i) \tag{27}$$

If (27) is substituted into the Lagrangian, we finally get:

$$\mathfrak{L} = \lambda \sum_{i \in p,u} \sum_{j \in p,u} \tau_i \tau_j \Phi(x_i)\Phi(x_j) - \frac{\alpha}{n_p} \sum_{i \in u,p} \sum_{j \in p} \tau_i \Phi(x_i)\Phi(x_j)$$

$$+ b \underbrace{\left(2\alpha - \sum_{i \in u} B_i - 2\sum_{i \in u} C_i\right)}_{0} + \sum_{i \in u} \xi_i \underbrace{\left(\frac{1}{n_u} - A_i - B_i - C_i\right)}_{0}$$

$$+ \sum_{i \in u} B_i + \sum_{i \in p,u} \sum_{j \in u} \tau_i B_j \Phi(x_i)\Phi(x_j) + 2\sum_{i \in p,u} \sum_{j \in u} \tau_i C_j \Phi(x_i) C_j \Phi(x_j) \tag{28}$$

Subject to:

$$A_i, B_i, C_i \geq 0 \tag{29}$$

$$2\alpha - \sum_{i \in u} B_i - 2 \sum_{i \in u} C_i = 0 \tag{30}$$

$$\frac{1}{n_u} - A_i - B_i - C_i = 0 \tag{31}$$

$$\tau_i = \begin{cases} \frac{\alpha}{2n_p} & , i \in p \\ -\frac{1}{2} B_i - C_i & , i \in u \end{cases} \tag{32}$$

We can apply the kernel trick to (28) and get a quadratic optimization problem with linear constraints (29-32):

$$\mathfrak{L}(\tau, B, C) = \lambda \sum_{i \in p,u} \sum_{j \in p,u} \tau_i \tau_j K(x_i, x_j) - \frac{\alpha}{n_p} \sum_{i \in u,p} \sum_{j \in p} \tau_i K(x_i, x_j)$$
$$+ \sum_{i \in u} B_i + \sum_{i \in p,u} \sum_{j \in u} \tau_i B_j K(x_i, x_j) + 2 \sum_{i \in p,u} \sum_{j \in u} \tau_i C_j K(x_i, x_j) \tag{33}$$

Similarly to the soft-margin SVM, each point with non-zero $\tau_i$ is a support vector. Since, all labeled positive examples have positive $\tau_i$, they are all support vectors. Because of large number of support vectors, this approach is too computationally demanding and time consuming, so we opt for optimization with SGD described in Appendix B.1.1.

## B.2 OC-SVM

We use scikit-learn implementation of OC-SVM in our work. We train all SVM models on features extracted with an encoder, which was trained as a part of an autoencoder only on positive data. We solve OC-SVM via dual problem, while PU-SVM is trained with gradient decent.

## B.3 OC-CNN

We implement OC-CNN according to the original paper (Oza and Patel, 2018).

## B.4 DROCC

For DROCC, we slightly modify the authors' implementation[1]. Originally, the adversarial search is performed in the input space. Because of that, negative examples are often original positive images with some noise. Instead, we search for negative examples in the feature space obtained from the output of the middle hidden layer of the network. In our experiments, we find that this small change improves ROC AUC.

## B.5 CSI

We use the official implementation of CSI [2].

## B.6 HRN

HRN (Hu et al., 2020) trains a neural classifier on the log-likelihood loss with a special regularization term $\|w\|^n$. The proposed term is referred as holistic regularization or H-regularization and essentially is an analogue of lasso or ridge regularization of a higher degree. The paper proposes to

---

[1]https://github.com/microsoft/EdgeML
[2]https://github.com/alinlab/CSI

set $n = 12$. Holistic regularization helps the model to remove feature bias and to prevent it from collapsing into a constant solution. We adapt the official implementation of HRN [3].

### B.7 PU METHODS

#### B.7.1 VPU

VPU (Chen et al., 2019) is based on variational inference that allows to estimate a variational upper bound of the KL-divergence between the real posterior distribution and its estimate, parameterized by a neural network. A major advantage of VPU is that variational upper bound can be computed without knowing prior probability $\alpha$, which most modern PU models rely on. Our implementation is based on the pseudo-algorithm provided in the original paper.

#### B.7.2 DRPU

We adapt official implementation of DRPU [4].

#### B.7.3 PAN

We use our implementation based on the original paper.

#### B.7.4 EN

EN (Elkan and Noto, 2008) is a classic PU algorithm. It consists of two steps. At the first step, a biased classifier is trained to naively distinguish positive data from unlabeled. At the second step, the output of the trained classifier is calibrated in order to make unbiased predictions.

#### B.7.5 DEDPUL

DEDPUL (Ivanov, 2019) is a two-stage algorithm. At the first stage, it trains a biased classifier to distinguish positive data from unlabeled and obtains predictions of this classifier for all examples. This is similar to the first stage of EN. At the second stage, it estimates the probability density functions of positive and unlabeled data in the space of predictions. Using the Bayes rule, these densities can estimate the posterior probability $p(x)$, whereas the prior probability $\alpha$ is chosen such that it equals the expected posteriors. DEDPUL achieves state-of-the-art performance in mixture proportion estimation and can improve accuracy of any PU algorithm.

We use the original implementation of DEDPUL[5] without any changes.

### B.8 TICE

In all our experiments, we use TIcE estimator of $\alpha$ for PU-SVM and PU-CNN models. We borrow TIcE implementation from DEDPUL repository[5].

## C DATASETS

**Twitter bots** In Appendix A.4, we additionally compare models on a dataset of Twitter accounts (Cresci et al., 2016; Rodríguez-Ruiz et al., 2020). This dataset provides information about around 7000 accounts, which are divided into genuine users and four different types of bots. On this dataset, we treat only one type of bot as known during training. On inference, we have four different options for negative data: the remaining three bot classes and all bot classes together. We acquire the data from the repository[6], which has differences in the number of classes and the account features with the dataset described in the original paper.

---

[3]https://github.com/morning-dews/HRN
[4]https://github.com/csnakajima/pu-learning
[5]https://github.com/dimonenka/DEDPUL
[6]https://botometer.osome.iu.edu/bot-repository/datasets.html

Table 13: Datasets' details for different settings in the main text

| Setting | Dataset | Pos class | Neg class | Pos lab | Pos unl | Neg unl | $\alpha$ |
|---|---|---|---|---|---|---|---|
| One-vs-all | CIFAR-10 | any class | all other classes | 2500 | 2500 | 2500 | 0.5 |
| Neg shift | CIFAR-10 | $\{0\}$ | rnd subset from other classes | 2500 | 2500 | 2500 | 0.5 |
| Pos modes | CIFAR-10 | subset from $\{0,1,8,9\}$ | all other classes | 2500 | 2500 | 2500 | 0.5 |
| Dependency on $|X_u|$ | CIFAR-10 | $\{0,1,8,9\}$ | all other classes | 2500 | vary | vary | 0.5 |
| Dependency on $\alpha$ | CIFAR-10 | $\{0,1,8,9\}$ | all other classes | 2500 | 2500 | 2500 | vary |
| – | Abnormal | Car | – | 576 | 573 | 86 | 0.87 |
| – | PenDigits | – | – | 2533 | 2509 | 110 | 0.97 |
| – | Dec. Reviews | – | – | 170 | 154 | 316 | 0.33 |
| – | SMS Spam | – | – | 1910 | 1937 | 609 | 0.76 |

Table 14: Datasets' details for new settings in Appendix

| Setting | Dataset | Pos class | Neg class | Pos lab | Pos unl | Neg unl | $\alpha$ |
|---|---|---|---|---|---|---|---|
| Neg shift | CIFAR-10 | $\{0\}$ or $\{2\}$ | see Table 9 or Table 3 | 2500 | 2500 | 2500 | 0.5 |
| Pos modes | CIFAR-10 | subset from $\{2,3,4,5,6,7\}$ | all other classes | 2500 | 2500 | 2500 | 0.5 |
| – | Twitter | Genuine | Social1 bot | 423 | 424 | 818 | 0.33 |

**Abnormal**   We use preprocessed Abnormal1001 dataset, where all images are resized to $200 \times 200$ resolution. Additionally for SVM-based models, we preprocess both CIFAR-10 and Abnormal1001 with encoders trained only on positive data. For CNN-based models, we normalize images with standard mean and std to apply ResNet.

**Text Datasets**   In order to preprocess text datasets, we remove stop words and punctuation, cast characters to lower case, and apply Snowball Stemmer from nltk package. Additionally, we use pretrained Google News embeddings.

**Train-test split**   For CIFAR-10 we use standard train-test split. Other datasets we split randomly into train and test in proportion $4 : 1$.

**Dataset details**   Tables 13 and 14 presents dataset details for each experimental setting. OC methods use only positive labeled data. PU methods also use unlabeled data but do not know which is positive and which is negative.

## D   HYPERPARAMETERS

Hyprameters for all models can be found in Tables 15, 16, 17, 18, 19, 20. First, we tune hyperparameters on one train-test split via grid search for all models except DROCC. Initial hyperparameters for DROCC we take from (Goyal et al., 2020). After that, we manually fine-tune hyperparameters in the neighborhood of the initial parameters found with grid search. For SVM-based methods we search on the following grid: kernel $\in \{rbf, linear\}$, $\nu \in \{0.1, 0.5, 0.9\}$, $\gamma \in \{0.01, 0.1, 1\}$, $\lambda \in \{0.01, 0.1, 1\}$, $lr \in \{10^{-3}, 10^{-4}\}$, num epochs $\in \{50, 100, 200\}$. For LSTM-based methods we search on the following grid: lstm dim $\in \{4, 16\}$, $\nu \in \{0.1, 0.5, 0.9\}$, batch size $\in \{128, 256\}$, $lr \in \{10^{-2}, 10^{-3}, 10^{-4}\}$, num epochs $\in \{50, 100, 200\}$. For CNN-based methods we search on

the following grid: $\gamma \in \{0.9, 1\}$, $lr \in \{10^{-3}, 10^{-4}\}$, num epochs $\in \{10, 20, 40\}$. For CSI-based methods we search on the following grid: $\lambda \in \{0.01, 0.1, 0.5\}$, $lr \in \{10^{-5}, 10^{-4}, 10^{-3}\}$, temperature$\in \{0.01, 0.05, 0.1, 0.5\}$, batch size$\in \{32, 64, 128\}$, $\gamma \in \{0.9, 0.96, 0.99\}$. For DRPU we search on the following grid: $\alpha \in \{0.001, 0.01, 0.1\}$, $lr \in \{10^{-4}, 10^{-3}\}$, $\gamma \in \{0.96, 0.99\}$, num epochs $\in \{10, 20, 50\}$. For PAN we search on the following grid: $\lambda \in \{0.001, 0.01, 0.1\}$, $lr \in \{10^{-4}, 10^{-3}\}$, $\gamma \in \{0.96, 0.99\}$, num epochs $\in \{10, 20, 50\}$.

Table 15: Hyperparameters for SVM-based models

| Hyperparameter | kernel | $\nu$ | $\gamma$ | $\lambda$ | num epochs | lr | lr decay |
|---|---|---|---|---|---|---|---|
| OC-SVM | linear | 0.5 | 2e-3 | – | – | – | – |
| PU-SVM | linear | – | 1 | 0.01 | 100 | 5e-3 | 0.995 |

Table 16: Hyperparameters for CNN-based models

| Hyperparameter | num epochs | lr | batch size | $\gamma$ |
|---|---|---|---|---|
| OC-CNN | 10 | 1e-4 | 256 | 1 |
| PU-CNN | 10 | 1e-4 | 256 | 1 |

Table 17: Hyperparameters for LSTM-based models

| Hyperparameter | num epochs | lr | batch size | $\nu$ | lstm dim |
|---|---|---|---|---|---|
| OC-LSTM | 200 | 1e-4 | 128 | 0.5 | 16 |
| PU-LSTM | 100 | 5e-3 | 256 | 0.5 | 16 |

Table 18: Hyperparameters for state-of-the-art PU models

| Hyperparameter | $\alpha$ | $\lambda$ | num epochs | lr | $\gamma$ |
|---|---|---|---|---|---|
| DRPU | 0.001 | – | 60 | 1e-3 | 0.99 |
| PAN | – | 0.001 | 20 | 1e-3 | 0.99 |

Table 19: DROCC Hyperparameters

| Model | DROCC | | | | | | | | | | | | PU-DROCC |
|---|---|---|---|---|---|---|---|---|---|---|---|---|---|
| Pos data | 0 | 1 | 2 | 3 | 4 | 5 | 6 | 7 | 8 | 9 | vehicles | Abnormal | All |
| $\lambda$ | 0.5 | 0.5 | 0.5 | 0.5 | 0.5 | 0.2 | 0.5 | 0.5 | 0.5 | 0.5 | 0.5 | 0.5 | 0.5 |
| radius | 24 | 24 | 32 | 28 | 32 | 36 | 32 | 28 | 28 | 28 | 28 | 30 | 2 |
| $\gamma$ | 1.5 | 1.1 | 1.5 | 1.1 | 1.5 | 1.5 | 1.5 | 1.1 | 1.1 | 1.1 | 1.1 | 1.1 | 2 |
| learning rate | 1e-3 | 1e-3 | 1e-3 | 1e-3 | 1e-3 | 5e-3 | 1e-3 | 1e-3 | 1e-3 | 1e-3 | 1e-3 | 1e-3 | 5e-4 |
| ascent step size | 1e-2 | 1e-2 | 1e-2 | 1e-3 | 1e-4 | 1e-3 | 1e-2 | 1e-2 | 1e-2 | 1e-2 | 1e-3 | 1e-3 | 1e-5 |
| ascent num steps | 40 | 60 | 60 | 60 | 20 | 40 | 60 | 50 | 50 | 50 | 60 | 50 | 10 |
| $\gamma_{lr}$ | 1 | 1 | 1 | 1 | 1 | 1 | 1 | 1 | 1 | 1 | 1 | 0.99 | 0.96 |
| num epochs | 30 | 30 | 30 | 30 | 30 | 30 | 30 | 30 | 30 | 30 | 30 | 15 | 20 |
| batch size | 128 | 128 | 128 | 128 | 128 | 128 | 128 | 128 | 128 | 128 | 128 | 128 | 256 |

Table 20: Hyperparameters for CSI-based models

| model | Parameter | 0 | 1 | 2 | 3 | 4 | 5 | 6 | 7 | 8 | 9 | vehicles | Abnormal |
|---|---|---|---|---|---|---|---|---|---|---|---|---|---|
| CSI | $\lambda$ | 0.1 | 0.1 | 0.1 | 0.1 | 0.1 | 0.1 | 0.1 | 0.1 | 0.5 | 0.5 | 0.1 | 0.1 |
| | batch size | 64 | 32 | 32 | 32 | 32 | 32 | 32 | 32 | 32 | 32 | 64 | 64 |
| | temp | 0.5 | 0.5 | 0.5 | 0.5 | 0.5 | 0.5 | 0.5 | 0.5 | 0.1 | 0.5 | 0.5 | 0.1 |
| | $\gamma$ | 0.99 | 0.99 | 0.99 | 0.99 | 0.99 | 0.99 | 0.99 | 0.99 | 0.99 | 0.99 | 0.99 | 0.99 |
| | $lr$ | 1e-4 | 1e-3 | 1e-3 | 1e-3 | 1e-4 | 1e-3 | 1e-3 | 1e-3 | 1e-3 | 1e-3 | 1e-4 | 1e-3 |
| PU-CSI | $\lambda$ | 0.1 | 0.1 | 0.1 | 0.1 | 0.1 | 0.1 | 0.1 | 0.1 | 0.5 | 0.5 | 0.1 | 0.1 |
| | batch size | 64 | 32 | 64 | 32 | 32 | 32 | 32 | 32 | 32 | 32 | 96 | 64 |
| | batch size unl | 16 | 16 | 16 | 16 | 16 | 16 | 16 | 16 | 16 | 16 | 32 | 32 |
| | temp | 0.5 | 0.5 | 0.5 | 0.5 | 0.5 | 0.5 | 0.5 | 0.5 | 0.1 | 0.5 | 0.07 | 0.01 |
| | $\gamma$ | 0.99 | 0.99 | 0.99 | 0.99 | 0.99 | 0.99 | 0.99 | 0.99 | 0.99 | 0.99 | 0.99 | 0.99 |
| | $lr$ | 1e-3 | 1e-3 | 1e-3 | 1e-3 | 1e-4 | 1e-3 | 1e-3 | 1e-3 | 1e-3 | 1e-3 | 5e-3 | 1e-4 |

