# OpenReview forum: "Improving State-of-the-Art in One-Class Classification by Leveraging Unlabeled Data"
_ICLR.cc/2022/Conference — ICLR 2022 Submitted_

### Official Review · Reviewer_kMPP · 2021-10-31

**Correctness:** 3
**Technical Novelty And Significance:** 2
**Empirical Novelty And Significance:** 2
**Recommendation:** 3
**Confidence:** 3

**Main Review:**

positives:
- The authors study a critical problem of leveraging unlabeled data to improve the performance of one-class classification tasks. The proposed PU modification of each one-class classification approach is easy to complete. Experiments show the improvements of the proposed approaches.

negatives:
- Although the experiments show the effectiveness of using PU loss in the classical one-class SVM, there lack theoretical analysis of the proposed approach, such as the consistency or the convergence rate.
- Section 3 lacks descriptions of each PU modified algorithm. The details of each method are unclear.
- The novelty seems limited. The proposed methods are straightforward combinations of one-class classification algorithms and PU loss. The authors report the experimental results on different PU modified algorithms, but it is still short of new insights for using unlabeled data in the one-class classification tasks.

**Summary Of The Paper:**

This work empirically investigates using the PU loss in Positive-Unlabeled learning in the one-class classification task. The authors propose several PU modifications of the original classification algorithms that can leverage the unlabeled data. Experiments show improvements after the modifications.

**Summary Of The Review:**

The authors propose several modified one-class classification algorithms, where part of the original loss is replayed by the PU loss in Positive-Unlabeled learning. Although the experiments show the effectiveness, there lack theoretical analysis of the proposed approaches.

---

### Official Review · Reviewer_99kf · 2021-11-02

**Correctness:** 3
**Technical Novelty And Significance:** 2
**Empirical Novelty And Significance:** 2
**Recommendation:** 5
**Confidence:** 3

**Main Review:**

Pros:
1. The paper conducts a good set of experiments with the goal of identifying cases when unlabeled data might not help

Cons:
1. Results are not surprising, lacks fresh insights, does not present concrete actionable steps
2. Alternate semi-supervised ways of using the unlabeled data (such as label propagation) have not been discussed.


Main Comments

1. Overall, the approach parallels semi-supervised algorithms where often adding unlabeled data helps and sometimes does not. This is well-known in general. The paper focuses on a specific class of algorithms to demonstrate this.

2. The paper tries to construct a few pathological cases (by setting up the data in certain manner) to bring the issues of introducing unlabeled data into focus, but since the results do not change dramatically (PU still wins in most cases), the examples look weak and not very illustrative.

3. The most important contribution would have been Section 5.6. However, this section is rather small. It would be better to relate \alpha to results discussed in (say) Section 5.2 in the context of specific datasets. I.e., some explanation along the lines of "Dataset X has \alpha = 0.10 and hence we see PU performing better/worse than OC and if we set \alpha=0.05, then we can achieve the opposite of this behavior..."

4. The paper's conclusion is rather fuzzy in the end (true \alpha probably is not known)... It would be good to list out concrete steps to identify dataset characteristics (such as unreliability of the unlabeled data), and offer prescriptions.

5. It would also be good to discuss how we can discard data that looks obviously unreliable and retain what is reliable.

**Summary Of The Paper:**

The paper suggests ways to extend purely supervised one-class classifier algorithms using techniques borrowed from PU algorithms in order to make use of unlabeled instances. The paper also presents situations where the unlabeled data (in addition to positive-labeled data) might not be helpful.

**Summary Of The Review:**

The paper attempts to address an issue of fundamental interest but falls short. The experimental results might still be useful to researchers.

---

### Official Review · Reviewer_R2RK · 2021-11-02

**Correctness:** 4
**Technical Novelty And Significance:** 1
**Empirical Novelty And Significance:** 2
**Recommendation:** 5
**Confidence:** 4

**Main Review:**

The empirical motivation provided in the paper clearly demonstrate the need for such study. By considering state-of-the-art OC algorithms, the authors perform PU modifications ranging from simply replacing the OC part with an existing PU part (in the case of OC-LSTM) or introduce appropriate  data into the unlabeled data part during model training. Such simple modifications is shown to have good have good empirical support for claiming PU modifications are better in 1-All scenario and random negative distributional shift. However, as the experiment reveals, non-random negative shifts still affect PU. So the primary question remains: even if distributional shifts are detected using Mann-Whitney test, how to know if this is a random or non-random shift? Furthermore, how to determine if unlabeled data is imbalanced and contain mostly positive data? Here too, the empirical results shown in table 3 shows mixed results. Surprisingly, the opposite seems true when sequential data is considered when the dataset with fewer positive examples exists. This is not clearly explained in the paper.

Another primary result in the paper is on the use of statistical tests to identify unreliability in unlabeled data. Though the authors mention 3 extremely important questions that provide recommendations on the methodology to be used, the test falls short in addressing when only a few unlabeled data is available. The primary question is how much is few? This may depend on variations in the data, model complexity etc. Unfortunately, the paper does not address this primary questions.

**Summary Of The Paper:**

The paper describes an empirical study performed on one-class (OC) and positive-unlabeled (PU) learning  settings. Here, the authors compare the OC and PU versions of multiple existing models under an un-reliable unlabeled data scenarios (such as scarcity and distributional shift of unlabeled data)  and provide practical recommendations to use PU algorithms, or modified OC algorithms for PU appropriately depending on data properties so as to leverage the superior classification efficiency of PU methods typically observed over reliable unlabeled data.

**Summary Of The Review:**

As the paper focussed on empirical evaluation of existing mechanism for providing practical recommendations, the paper's brings out the need for understanding unlabeled data distributions before blind application of OC or PU methods. However, there still exists key unanswered questions needed for practitioners to incorporate the recommendations in practical settings.

---

### Official Review · Reviewer_9LXT · 2021-11-03

**Correctness:** 2
**Technical Novelty And Significance:** 2
**Empirical Novelty And Significance:** 2
**Recommendation:** 3
**Confidence:** 4

**Main Review:**

Strengths.
- The study problem, exploring large unlabeled data with some positive examples, is important
- The idea of extending OCC models to PU learning settings is interesting
- The empirical results show the effectiveness of adapted OCC models in some PU settings.

Weaknesses.
- PU-OCC models have advantages over PU models only in some extreme cases, such as when the unlabeled data contains mostly positive examples, when there are distribution shifts, etc. These are not surprising as OCC models are specifically designed for such settings. Further, for those unusual settings, there are more one-class classification or anomaly detection problems than PU learning. From this perspective, these settings are somehow ill-posed
- The empirical results are not convincing enough, as most of the empirical justification are based on one single dataset CIFAR-10.
- The performance of different methods is dependent heavily on the proportion of positive examples in the unlabeled data. The parameter \alpha is pre-fixed for each specific dataset. I wonder what would be the empirical observations when the pre-fixed parameter \alpha is changed in different data.
- The PU-OCC models are developed based on existing studies; I cannot find any major technical novelties

**Summary Of The Paper:**

The work presents an empirical study of extending one-class (OC) classification to PU learning in order to leverage unlabeled data more effectively, especially when the unlabeled data contains only a small proportion of positive samples. Five OC classification models are used and evaluated on a few classification benchmarks.

**Summary Of The Review:**

The studied problem is important, but there are a number of major issues in the main claims, the empirical justification and technical novelty. So, I do not recommend it for publication at ICLR.

---

### Decision · Program_Chairs · 2022-01-20

**Decision:**

Reject

**Comment:**

This paper provides empirical results for one-class classification problems.
The studied problem is important and the reviewers admire the challenge of this paper.
However, the empirical results are not still insightful enough to provide practical recommendations.

Some of the questions raised by the reviewers could potentially be answered by the authors, but we did not receive any feedback unfortunately.

Given that there is essentially no technical novelty in this paper, it cannot be accepted for ICLR.